# Resilience amid Adversity: A Qualitative Narrative Study of Childhood Sexual Abuse Among Bangladeshi Transgender Individuals

**DOI:** 10.3390/ijerph22040615

**Published:** 2025-04-15

**Authors:** Sanzida Yeasmin, Jennifer J. Infanti

**Affiliations:** Department of Public Health and Nursing, NTNU-Norwegian University of Science and Technology, 7034 Trondheim, Norway; sanzidayeasmin37@gmail.com

**Keywords:** childhood sexual abuse, resilience, transgender, trauma, advocacy, Bangladesh, coping mechanisms, abuse disclosure

## Abstract

Childhood sexual abuse (CSA) is a prevalent form of violence against children, associated with profound negative impacts on survivors’ health and well-being. In Bangladesh, sociocultural and economic barriers hinder CSA prevention and intervention, particularly for vulnerable populations such as transgender individuals. This study explores the experiences of CSA among Bangladeshi transgender individuals, focusing on their coping mechanisms and resilience-building strategies. A qualitative narrative approach was employed, using life story-telling interviews with four transgender CSA survivors, and data were analyzed thematically. The findings reveal significant challenges, including social exclusion, humiliation, and lack of family support. Participants reported experiencing physical injuries, mental trauma, and loss of trust due to CSA, while the conservative nature of Bangladeshi society often prevented them from disclosing their abuse or seeking healthcare. These barriers exacerbate health inequalities among gender-minority children. Despite these adversities, participants described engaging in resilience-building strategies such as self-care, personal development, advocacy, and efforts to derive meaning from their experiences, while highlighting the perceived importance of family support. This study provides insights into the unique experiences of transgender CSA survivors in Bangladesh and calls for efforts towards gender-inclusive education, mental health support, and further research to address CSA-related adversity, encourage disclosure, and promote health equity for gender-minority groups.

## 1. Introduction

Traumatic early-life experiences, such as childhood sexual abuse (CSA), can have profound and long-lasting negative effects on health and well-being [1]. The World Health Organization (WHO) defines CSA as the involvement of a child in sexual activity that they do not fully comprehend, cannot give informed consent to, or are not developmentally prepared for [2]. Such experiences violate societal laws or social taboos. CSA is associated with a wide range of adverse health consequences. A recent systematic review documented that survivors often experience mental health issues, including depression, anxiety, and post-traumatic stress disorder (PTSD) [3]. Physical health problems, such as genital injuries, urinary tract infections, and early pregnancies, are also prevalent [3,4]. Additionally, survivors face an increased risk of substance misuse, sexualized behaviors, and sexually transmitted infections, including HIV/AIDS [3,4]. CSA has been identified globally across all levels of society, in virtually every region and continent, including in Bangladesh [5,6].

Bangladesh, a South Asian country, is a young nation established in 1971 as the People’s Republic of Bangladesh after Bengali East Pakistan seceded from (West) Pakistan [7]. It is one of the most densely inhabited countries internationally, with an anticipated 171 million people in 2021 [7]. Dhaka is the country’s capital and largest city, and Bangla (or Bengali) is the primary language spoken by 98% of the population [7]. Bangladesh has achieved several of the United Nations’ Millennium Development Goals (MDGs), drastically narrowing the poverty gap from 56.7 per cent in 1991–1992 to 24.8 per cent in 2015 [8]. It was ranked 136 out of 189 nations in 2017 as one of the countries with a medium level of human development [8]. Despite this development, one out of every four individuals remains poor, with 13% living in extreme poverty [8].

In addition to broader development challenges, Bangladesh faces significant public health concerns, including widespread CSA, with studies reporting frequent incidents of sexual violence and coerced intercourse [6,9,10]. Factors such as inadequate sexual education, traditional family and kinship systems, weak law enforcement, and limited public awareness contribute to the persistence and apparent rise in CSA cases [11]. Legal frameworks do exist to prosecute CSA in Bangladesh. For example, the ‘Nari o Shishu Nirjaton Daman Ain 2020’ criminalizes sexual oppression, prescribing 3 to 10 years of rigorous imprisonment and a fine [12]. The law defines sexual oppression as any act in which an individual illegally touches the sexual or other organs of a woman or child to satisfy sexual desire, using any part of the body or an object [13]. In addition, Bangladesh’s penal law defines five forms of rape, including statutory rape where the victim is under 14, regardless of consent [14]. The Domestic Violence (Prevention and Protection) Act 2010 further recognizes sexual harassment by a family member as a form of domestic violence, allowing survivors to seek legal remedies [15]. Based on these legal provisions, survivors can file complaints against CSA and seek justice. Yet, many CSA incidents remain unreported due to societal taboos, stigma, and limited awareness [5,16]. Survivors often experience shame and fear of social rejection or humiliation, which hinders both disclosure and access to justice [5,17].

The term ‘transgender’ broadly refers to individuals whose gender identity, expression, or behavior differs from conventional expectations of their assigned sex at birth [18]. Gender identity refers to an individual’s internal sense of their own gender, while gender expression involves outward presentations such as clothing, hairstyle, and behavior [18]. Transgender individuals in Bangladesh are particularly vulnerable to CSA due to compounded challenges such as prejudice, discrimination, and legal barriers to accessing employment and healthcare [19,20].

Bangladeshi social values and religious beliefs influence gender identification and do not accept individuals who do not conform to the male-female gender pattern [21,22]. Fundamental civil and human rights like inherent property rights and marriage are withheld from these people since carnal intercourse is “against the order of nature”, as declared in section 377 of the Bangladesh penal code [19]. In addition, as the country has no anti-discrimination law that specifically protects sexual minorities, and there is no policy recognizing the diversity of gender identity [21], transgender individuals live on the margins of society, with little socio-political space to live a life of dignity [23]. Although the government in Bangladesh acknowledged transgender individuals as a ‘third gender’ in 2013 [21], and Bangladeshi Constitution protects the rights of individuals by stating that the state shall not discriminate against its citizens on the grounds of religion, race, caste or sex, due to a lack of laws recognizing their status, transgender people have been excluded from the fundamental rights associated with citizenship [21] and face discrimination from social, cultural, legal and healthcare institutions, resulting in diminished self-esteem and a sense of social responsibility [23,24]. However, these people confront socio-cultural and religious barriers and discrimination, perhaps more so than legal issues [19]. Research has consistently shown that transgender individuals face an elevated risk of sexual abuse [25,26]. A study links childhood sexual abuse (CSA) in transgender women to homelessness, stress, and a higher risk of HIV, with health impacts similar to those in cisgender individuals [27].

Taken together, these structural vulnerabilities mean that CSA may have particularly harmful effects on the physical and mental health of transgender individuals in Bangladesh. It may also increase their susceptibility to gender-based violence (GBV) and contribute to broader health disparities. Exploring their lived experiences and coping strategies offers critical insights into social marginalization, transgender rights, and GBV in this context. Such research can inform targeted interventions, legal protections, and improved support systems tailored to their needs. In addition, these findings have potential relevance for other low- and middle-income countries with similar socio-cultural environments. Addressing these gaps is an important step toward developing effective prevention and response strategies for transgender survivors of CSA.

However, studies focusing on CSA among transgender individuals in Bangladesh remain limited. Despite their increased vulnerability, the lived experiences of transgender CSA survivors are largely undocumented. Few studies capture their narratives in depth, beyond trauma, to highlight strengths and health-promoting factors within this community.

In Bangladesh, poverty, lack of education, and social awareness contribute to CSA and human rights violations against gender minority communities, impacting their overall well-being. Addressing resilience among CSA survivors can support recovery from trauma-related conditions such as PTSD, anxiety disorders, and depression while promoting overall health and well-being more broadly [28]. Resilience means a positive adaptation after stressful conditions, i.e., the potential of an individual to successfully adjust to change, resist the negative influence of stressors, and prevent substantial dysfunction [28]. Resilient individuals often demonstrate better psychological outcomes and enduring positive effects [29]. It also appears that the greater a person’s resilience, the lower their susceptibility to illness and likelihood of developing it, even despite chronic illness [28]. The strength to bounce back from adversity and go back to “normal” or “healthy” states after experiencing trauma, accident, tragedy, or illness makes resilience a crucial concept to study in the field of medicine and health [28]. Yet, existing research lacks resilience, health promotion, and support resources. Notably, none explore how transgender women make sense of CSA experiences in adulthood and the coping strategies they used to overcome trauma and challenges in their lives.

Given these considerations, this study explores the life stories of resilient Bangladeshi transgender women who have survived CSA, with a focus on their coping and resilience-building strategies. By documenting their experiences, the study seeks to contribute to a deeper understanding of how survivors navigate adversity and identify pathways to their well-being.

## 2. Materials and Methods

The authors designed the study to explore the life experiences and resilience strategies of Bangladeshi transgender individuals who are survivors of CSA. As researchers with shared interests in global health, gender-based violence, resilience, and child welfare, they acknowledge that their academic and personal backgrounds have shaped the development and interpretation of this study. Drawing on expertise in medical anthropology and narrative research in South Asia, they approach the work through an ethical, trauma-informed, and culturally sensitive lens.

A qualitative research approach using narrative life story interviews was chosen to gather data. This method connects personal experiences to social and cultural contexts, providing insights into significant life challenges and triumphs [30,31].

To minimize potential emotional distress, only healthy, consenting adult transgender women who had survived CSA and were willing to participate were included. The final sample consisted of four transgender women aged between 25 and 40 years, who self-identified as healthy, coping, and eager to share their stories.

Participants were recruited by the first author through social media outreach, collaboration with non-governmental organizations (NGOs) working on transgender rights, and networking facilitated by the goodwill ambassador of one of the NGOs in Dhaka, Bangladesh. They were provided with a formal study information sheet and a recruitment poster to distribute within their networks. Potential volunteers who met the inclusion criteria were invited to contact the researcher (first author) for further details. Five potential participants initially expressed interest and attended individual online meetings and received written study information. After online introduction meetings, four participants proceeded to the final interviews conducted by the first author. Interviews took place in private locations chosen by participants in Dhaka, Bangladesh.

An existing life story interview format was used with ten open-ended questions to guide the interviews [32]. Prior to data collection, two mock interviews were conducted to ensure the relevance and comprehensibility of the guide and to avoid direct questions that might cause distress. At the beginning of each interview, participants were informed about the study’s purpose, research goals, and ethical considerations. They were explicitly made aware of their rights to decline to answer any questions or withdraw from the study at any time without explanation. Efforts were made to maintain a comfortable and respectful interview environment. Though the interview guide was written in English, the first author preferred to conduct the interviews in ‘Bangla’ as all the participants were Bangladeshi. It was assumed that they would feel most comfortable explaining their experiences and expressing their feelings in their native language. With the participants’ permission, all four interviews were audio-recorded using a digital recorder. To protect their identities, special care was taken during transcription to de-identify the information. The first author manually transcribed the recordings to avoid technical errors, after which all audio files were deleted. The transcripts were then translated into English. Each participant received a copy of their transcript for review and feedback, ensuring the research was clear and meaningful. This also allowed them to provide additional insights, enriching the analysis [33].

The life stories shared during the interviews were condensed into concise biographical summaries. These were created by selecting important parts from each interview that can answer the study’s research aims and highlighted key topics, including childhood experiences, challenges, coping strategies, and future plans. To protect participants’ privacy, pseudonyms were assigned. Subsequently, the written summary of life stories was presented to the research participants for their review, reflection, and correction before the analysis began.

A thorough thematic analysis was conducted by the first author to identify, examine, and interpret patterns in the data [34]. This analysis enabled the coding and categorization of the data into themes that captured meaningful aspects of the participants’ narratives in relation to the research objective and reflected structured patterns within the dataset [34,35]. Coding involved organizing the data into meaningful units and served as the first step in identifying potential themes [34]. In this study, the first author generated codes from either a single sentence or, in some cases, longer paragraphs in the interview transcripts that contained information relevant to the research objective (for example, a sentence expressing an idea related to resilience), assigning each code a descriptive label. An inductive approach was primarily used, meaning the data were coded without being fitted into a pre-existing coding framework or theory, allowing themes to emerge directly from the participants’ accounts [34].

After coding all transcripts, the first author grouped the codes into potential themes and compiled the relevant coded data extracts under each theme [34]. These were reviewed in collaboration with the co-author to ensure they accurately reflected the content and meaning of the full dataset [35]. The first author then re-examined the interview transcripts to identify any important information that may have been initially overlooked, incorporating new codes and removing data that appeared irrelevant or redundant. Potential themes and associated codes were then compiled and, where appropriate, organized into sub-themes to provide structure to more complex themes. The data categories within each theme were reviewed to assess their coherence, relevance, and distinction [34].

The research findings were presented under these themes through summary and interpretive analysis. Finally, the themes were compared with findings from existing literature. Throughout the process of presenting the data, the first author prioritised coherence and consistency in the narrative representation of participants’ experiences.

The authors adhered to WHO ethical guidelines to ensure participants’ confidentiality, safety, and dignity [36]. Interviews were conducted in private settings, pseudonyms were used, participants were informed about audio recording and data storage, and all recordings were securely handled. Participants’ rights to refuse questions or withdraw consent were respected, ensuring a supportive and ethical research environment [36].

## 3. Results

This section presents summaries of the life stories of participants who experienced childhood sexual abuse (CSA) and demonstrated resilience in overcoming significant challenges. These summaries provide readers with a foundation for understanding the core elements of their narratives [37]. Specifically, the authors present key research findings by offering brief biographical accounts drawn from each participant’s longer life story. To create these summaries, passages were selected from the interviews that reflected the main areas of the interview guide: overall childhood experiences (including CSA), significant personal challenges, coping strategies, future aspirations, and concluding reflections.

The participants have been assigned pseudonyms—Dilshad, Lavanya, Bivabori, and Maya—to protect their identities. These personal accounts, shared largely in the participants’ own words, offer insight into their lived experiences, highlighting the hardships they endured and the strategies they used to cope. The summaries aim to preserve the participants’ voices without additional interpretation or analysis, providing a deeper understanding of their individual journeys toward resilience and advocacy.

### 3.1. Dilshad’s Story


*My name is Dilshad. I am a Bangladeshi transgender woman who has endured untold hardships, prejudice, ridicule, poverty, and violence since childhood. However, through my efforts and ongoing self-improvement, I am trying to prove myself as a successful person and fighting for the welfare of minority people worldwide.*



*Being raped the whole night at an early age was the darkest experience for me in life. Besides intolerable pain and bleeding, I was mentally traumatized. I had no idea what sexual abuse was, and I didn’t know how to complain or protest because I didn’t know the right words, nor was I able to disclose this abuse and seek a physician’s help because of conservative societal beliefs. As a boy, it was surprising and challenging to admit to being sexually abused by another male in childhood, but it was even more shocking to experience such behavior from someone I considered a very pious man.*



*Having endured multiple cases of abuse since childhood, I have also suffered thousands of other types of traumas–being excluded, existential crises, discrimination, dehumanization, extreme poverty, feeling marginalized in our LGBTQ community, and so on. In our society, it’s not easy to be a transgender woman, and it always broke my heart when people were cruel to me.*



*Despite confronting challenges concerning to my gender identity, including concern about my education and employment prospects, I continued to work hard. I fought for the positive outcomes, finished my studies, got a job, and sought professional help to better understand and embrace my identity. Although the path was challenging, I found it rewarding, as it allowed me to feel comfortable in my own skin.*



*However, revealing my true gender identity cost me a lot. I lost many of my near and dear ones, but I was strong enough to hold my identity. I got this strength by reading books and in continuous self-improvment. These books gave me the strength to keep going and motivated me to work towards improving myself and helping others. Besides that, I focused on self-improvement to cope with trauma and sought connections with like-minded individuals nationally and internationally, which has been crucial for my well-being.*



*Now, I work for the welfare of minority people and want to make the world a better place for them. It is not always easy, especially in a society dominated by conservative people, where violence against transgender people is a real fear and reality. But I believe that I have a purpose in life and am committed to working towards it, no matter if obstacles come my way.*


### 3.2. Lavanya’s Story


*I am Lavanya. I always prioritize family bonding and human relationships over everything else. Even though I became a social outcast because of my gender transformation, I am uncompromising in taking responsibility for my family, society, and humanity.*



*My childhood was not pleasant compared with that of others; it was full of torture, physical harassment, fear, self-hatred, and a series of sexually violent experiences.*



*With so many individuals, all of whom I thought were my near and dear ones, abusing me so much, I had difficulty deciding whom to trust. I struggled to understand what was happening to me. I spent nights in fear and pain. I started hating myself and wanted to escape from this suffocating, abusive environment again and again. My biggest problem was that I could not tell anyone about it, not even my mother, because she wouldn’t believe me, she would rather say, he is your uncle, how can he do this sorts of things to you? Neither could I go to a physician despite serious injuries, as they wouldn’t believe me. As a boy child with no female parts, it was hard for me to understand why men would abuse me in this way. Since I was from a remote village, I didn’t know much, but I think my girly behavior was the reason behind all those abuses. Yes, from my early childhood, I used to do girly things; I enjoyed girly games, and my tone of voice and attitude were also girly. I suffered greatly because of that. However, I never felt ashamed of being that way and dreamt of being a woman since I was a kid.*



*I always wanted to get through my traumatic experiences, move on with my life, and become a transwoman. With the support of my parents, and through my initiatives, I managed. I underwent gender transformation and began to thrive. Coming out as a transwoman and gaining success, I also gathered the courage I needed to be resilient. Although life as a transwoman can be challenging in our society, I face these challenges with confidence, completing my education and working for a multinational company, while also pursuing a passion for modeling. Most importantly, I use my experiences to help others, engaging in charity work for minority people, and planning to do more for underprivileged students and older adults.*



*I believe that peer and social support is needed for one’s well-being, but self-love and self-reliance are things that I value highly in human life. Besides being resilient and maintaining my dignity, I am now turning my tragedies into strengths, holding myself firm, and protesting all kinds of adversities.*


### 3.3. Bivabori’s Story


*My name is Bivabori, and I am a Bangladeshi transgender woman. I grew up in a traditional, non-supportive Bangladeshi family, where boys being girly or transgender was not appreciated; instead, bullying was practiced frequently. My childhood was a painful era filled with neglect, humiliation, and bullying, and I wouldn’t ever want to return to that time.*



*I’ve always found it difficult to handle life’s problems without basic family support. No one in my family liked me because I was feminine and enjoyed things that were considered girly, such as cooking, painting my nails, and applying henna. Besides my family, I was also rejected by society. I was kicked out of the mosque during a religious event because of my feminine behavior. It was indeed a frightening experience for me as a child. But that wasn’t the only bad incident that happened to me during childhood. I was also sexually abused at a young age, which affected me greatly. I felt suffocated because I couldn’t share it with anyone, not even my mother, because discussing sexuality is such a taboo in our society. When I managed to find the courage to complain to my family, nothing was done, and I was scolded instead. My family didn’t support me through these situations, nor did they accept me as myself. I felt alone and had no one to turn to during difficult times. No matter how often similar things happened afterwards, I was too scared to talk about them, which made me feel more traumatized.*



*Being transgender in Bangladeshi society and having childhood trauma and adversities made it challenging to complete my studies and get a job. However, recognizing the importance of financial independence for self-confidence and autonomy, I fought through adversity to excel in my studies. Drawing inspiration from the transgender community, I stayed motivated. After completing my studies and getting employment, I felt a significant change in my life. I got the courage to reveal my true gender identity to my family. Also I gained the confidence to face life’s challenges. Encouraged by the positive atmosphere at work, I now embrace good vibes and remain deeply motivated.*



*As a transwoman, I empathize with the struggles of fellow minorities. That’s why I am committed to supporting transgender individuals by mentoring, providing training for job interviews, and offering mental support to those in need. All of these brings me mental peace, and I aim to continue it. In addition to that, I feel the importance of social support for well-being.*



*My life’s theme is simple: to be honest with myself and continue working hard. I don’t think it’s a good idea to stop; rather, one should move with life for their well-being.*


### 3.4. Maya’s Story


*I’m Maya. Despite childhood obstacles, I have established myself in society. I view challenges as integral to life and now aim to help people of all backgrounds find well-being through reflection on my experiences.*



*Earlier in my childhood, the torture inflicted on me outweighed everything else. Even though I was biologically born a boy, my surroundings made me understand that I am transgender. I am different from others and can be humiliated because of my differences. I was socially excluded in school. Day after day, I was subjected to all kinds of abuse, and it seemed that no one would listen to me when I tried to speak up. Even when I told my mother and various other family members about the sexual abuse I had been going through, I was ignored. My mother, for some reason, tried to hide it, saying, “Nothing happened, nothing happened”, which exacerbated my suffering. My studies were also deteriorating.*



*I’ve been ambitious since my early ages to make a positive change in my life and for others. Overcoming obstacles, especially in completing my studies, was challenging. My mother’s motivation and support for my studies were most effective for me. In addition to the prescribed schoolbooks, I read many other works. I enriched my mind by reading about great authors. After years of continuous effort, I completed my higher education and now aim to pursue a PhD in gender studies. Besides being successful in life, I always prioritize well-being. I maintain a balance through healthy habits, meditation, and counseling. My interests in dance, poetry, and writing bring joy, while my passion for helping others transcends the boundaries of ethnicity, religion, or caste. During the COVID-19 pandemic, I provided support, participated in over 700 final rites, and offered counseling to those in need.*



*My childhood was symbolic of strength, and I continue to use the strength I’ve gained to inspire and help others. I believe that the most valuable thing in human life is respecting people as human beings and animals, and I always strive to be an enlightened and honest person. Working for the betterment of people is the supreme task, and I am proud to be doing my part.*


The narratives of the transgender women in this study reveal diverse pathways of resilience in coping with CSA, highlighting both personal and structural challenges. Their experiences underscore the pervasive impact of cultural stigma, lack of family support, and social exclusion, factors that hinder disclosure and limit access to essential support services. At the same time, the participants demonstrated post-traumatic growth (PTG) through resilience, advocacy, and community engagement. Many found strength in their identities, participated in activism, and developed coping strategies that fostered both personal healing and collective empowerment.

PTG, defined as “a positive psychological change experienced as a result of the struggle with trauma or highly challenging situations” [38], provides a useful lens through which to understand these transformations. Although PTG can coexist with negative psychological consequences, it represents a parallel process of growth. It may lead to positive changes in self-perception, interpersonal relationships, and life philosophy, including greater self-awareness, increased self-confidence, a deeper appreciation of life, and the discovery of new possibilities [38].

Despite the profound challenges they faced, the participants’ stories illustrate resilience through self-improvement, self-love, and advocacy, offering valuable insights for future interventions aimed at promoting health equity among marginalized communities. These threads of resilience and PTG are taken up and further examined in Section 4, where we explore their broader implications for health systems, policy, and support strategies.

## 4. Discussion

To provide a structured understanding of the findings, the discussion is organised around the three overarching themes identified through thematic analysis of the narratives: gender identity and societal norms, trauma and resilience, and advocacy and social change for health promotion.

### 4.1. Gender Identity and Societal Norms

Religious and cultural values strongly influence societal perceptions of gender identity and sexual relationships in Bangladesh, contributing to the stigma and discrimination experienced by transgender individuals [39]. Conservative interpretations of religion—often portraying transgender people as sinners [22]—reinforce societal opposition and are used to justify their exclusion [39]. Gender non-conforming individuals are generally not accepted in mainstream Bangladeshi society and are instead subject to stigma, marginalization, and mistreatment [21,24]. In some respects, their situation is considerably more complex than that faced by transgender individuals in many Western societies [21]. Compared to the general population, transgender people in Bangladesh are more likely to experience stressful life events and are subjected to multiple forms of discrimination. Documented challenges include social exclusion, isolation, financial hardship, legal difficulties, and physical, sexual, and psychological abuse, all deeply rooted in prevailing societal and cultural beliefs [23,37,38].

This study illustrates how deeply ingrained societal norms can result in bullying, harassment, and social exclusion, often beginning in childhood [21,24]. The participants’ narratives reflect experiences of family rejection, social exclusion (in terms of education, practicing religion, and obtaining employment), and isolation, exacerbating their struggles with gender identity disclosure. People generally face various barriers in completing education and pursuing a career; however, the research findings show that transgender individuals in Bangladesh encounter additional and distinct challenges due to their gender identity. Compared to their cisgender peers, they endure more frequent bullying, humiliation, and mistreatment on a daily basis, often without any societal support, factors that significantly impact their self-esteem. A lack of basic empathy and support can exacerbate mental health difficulties and hinder recovery. Studies suggest that, compared to those who have not been bullied, victims of bullying face a range of adverse mental health, academic, and life outcomes [40].

The findings also reveal that most participants experienced a non-supportive family environment due to their gender non-conforming behaviors, which led to exclusion within their own families. This rejection intensified their suffering by exposing them to further bullying, humiliation, and abuse outside the family. Bangladeshi culture places a strong emphasis on family reputation and adherence to strict gender norms; as a result, a biological male displaying feminine traits or interests is often seen as dishonorable and a source of shame for the family [21]. This cultural stigma often prompts families to deny support to their gender non-conforming children.

While family support is known to be a protective factor against bullying and mental distress [41], in the Bangladeshi context, transgender individuals often struggle to access such support. This lack of familial acceptance can cause deep anxiety about disclosing one’s gender identity. In Bangladesh, transgender individuals frequently face considerable social consequences when expressing a gender identity outside of the traditional male-female binary [22]. Participants in this study shared similar concerns, expressing fears of humiliation, loss of loved ones, social exclusion, denial of basic rights, and additional suffering inflicted upon their families. Because of the anticipated discrimination and negative consequences associated with ‘coming out’, transgender individuals are often forced to choose between affirming their gender identity for personal integrity and avoiding the interpersonal costs of doing so [22].

The study findings reinforce a need for gender-inclusive policies, including school-based interventions such as teacher training programs to foster acceptance and reduce stigma. Furthermore, initiatives aimed at parental education could help promote familial acceptance of gender-diverse children, creating a more supportive environment for their development.

### 4.2. Trauma and Resilience

Violence against children is a matter of public health, human rights, and social concern, with potentially severe and long-lasting consequences [42]. It remains one of the most significant obstacles to protecting children’s rights and safety in Bangladesh [8]. Gender non-conforming children are particularly vulnerable, facing higher levels of violence compared to their cisgender peers, often beginning in early childhood [43]. Childhood abuse featured prominently in the life stories of all participants in this study. Among the emotional, physical, financial, and sexual abuse they experienced, childhood sexual abuse (CSA) was the most frequently reported.

According to a WHO estimate, globally, up to 150 million girls and 73 million boys under the age of 18 experience forced sexual contact or other forms of physical or sexual assault [42]. Although the WHO report does not specifically mention transgender individuals, all participants in this study reported experiencing CSA between the ages of 5 and 12. They described CSA as profoundly traumatizing, resulting in persistent PTSD-like symptoms, self-hatred, fear of intimacy, and a desire to escape abusive environments [43].

The study findings suggest that transgender individuals experienced CSA more frequently than their cisgender peers. Two participants attributed their abuse to their femininity, reflected in their behaviors, voices, attitudes, and clothing. One study also indicates that a biological male’s attraction to female attire, behaviors, or household tasks can contribute to sexual harassment in the Bangladeshi context [21]. Although sexual assault is known to be more prevalent among transgender people than among cisgender individuals [25], participants in this study experienced CSA before openly identifying as transgender women. This suggests that their gender non-conforming presentation may have contributed to their vulnerability.

While all CSA survivors suffer negative consequences, transgender survivors in Bangladesh face additional layers of suffering due to the country’s socio-cultural context. A lack of sexual education and self-defense training contributes to their inability to recognize, resist, or report CSA. In conservative Bangladesh, talking about sex is considered taboo, making sex education controversial and rarely discussed [44]. The participants’ narratives reveal their difficulty in understanding what had happened to them, particularly in making sense of why a man would engage in sexual acts with individuals assigned male at birth. This reflects a broader lack of knowledge regarding CSA and homosexuality, both of which are viewed in traditional Bangladeshi society as deviant and deeply distressing.

Although there is some indication that CSA is widespread in Bangladesh, obtaining accurate prevalence data remains difficult due to non-disclosure, largely driven by cultural stigma [6]. Many children still do not feel comfortable discussing CSA with parents or family members, as they do not perceive them as approachable or supportive in matters related to sexuality. Furthermore, both society and survivors may downplay or misclassify certain forms of CSA, such as touching or fondling of genitals or breasts, as minor, and therefore not worth formally reporting [5].

However, disclosure is known to benefit CSA survivors, with parental care and support being key to recovery [45]. Conversely, persistent parental neglect or absence in response to disclosure increases vulnerability to ongoing abuse and contributes to high rates of non-disclosure [45,46]. This study’s findings indicate that families often focused on the child’s gender non-conformity rather than the abuse itself. When participants reported assaults, they frequently encountered neglect or victim-blaming, as families perceived their femininity as a contributing factor. One study notes that current social practices in Bangladesh often aim to hide anything deemed “shameful”, including rape or sexual abuse [6].

Additionally, boys may be especially reluctant to disclose sexual abuse due to gender socialization, which emphasises sexual dominance, heteronormativity, and emotional suppression [47]. Being born male and abused by a man, several participants feared disbelief, stigma, or rejection, particularly given the strong societal rejection of homosexuality in Bangladesh. This fear contributed to their silence, illustrating how societal and cultural beliefs shape the non-disclosure of CSA and increase vulnerability to repeated abuse.

Although self-reporting is recognized as crucial for addressing violence against children, both within the Sustainable Development Goals (SDGs) and for individual wellbeing, links have also been established between CSA and future health risks, including HIV, STIs, and high-risk behaviors in adulthood [42,48]. Yet many participants in this study hesitated to disclose their abuse, even to healthcare providers. The WHO recommends that all child and adolescent CSA survivors receive timely medical care, including HIV post-exposure prophylaxis (within 72 h), treatment for STIs, emergency contraception, and psychosocial support [49]. In this study, two participants described injuries resulting from CSA but were reluctant to seek medical care due to their gender non-conformity, stigma, and fear of negligence.

A survey indicates that fewer than 10% of CSA survivors in low- and middle-income countries receive healthcare services, figures which vary significantly by country and gender, with boys being less likely to access services than girls [49]. These findings suggest that in Bangladesh, social and cultural beliefs may prevent children from accessing even their most basic rights, including medical treatment for sexual abuse or psychological support. This has serious implications for their short- and long-term health. These barriers contribute to significant health risks and exacerbate health disparities among gender non-conforming children.

CSA is a critical issue among transgender individuals in Bangladesh, with survivors facing significant physical and emotional trauma [43]. This study shows that the impact of CSA among them can be significant due to conservative social and cultural beliefs. Additional research and intervention are required to encourage parents to provide sexual education to children, provide a safe place for reporting CSA, ensure the availability of healthcare services, and support CSA survivor kids regardless of their gender identity.

Despite the challenges mentioned in terms of being transgender and experiences of CSA, the participants demonstrated resilience by dedicating substantial time and effort to various coping strategies, such as self-love, self-improvement, turning tragedy into strengths, and engagement in creative activities. They desired resiliency and set out to adapt to hardship; in some ways, this seemed deliberate, while in others, it came as a result of circumstances. These findings align with research emphasizing the role of self-awareness and conscious emotional regulation in fostering resilience among individuals facing societal rejection [50]. The importance of social support—particularly through peers and community networks—emerged as a critical factor in navigating trauma and fostering inner motivation, optimism, and self-confidence [41].

Self-love fosters self-worth, confidence, and self-acceptance regarding sexual identity, protecting against LGBTQ-related prejudice and rejection [50]. Self-love and care may aid in overcoming adversity or at least provide the fortitude to combat negativity and inhibit risky behaviors such as suicide. Perhaps it exhibits an inner confidence in people, which enables them to endure adversity and move forward to live lives of their choosing without feeling deprived or humiliated. This was seen in the transgender research participants in this study, who found self-love as a means of resiliency. It suggests that individuals may move on and let go of negative memories if they concentrate on their well-being and act accordingly.

In addition to surviving and thriving, they consistently worked on self-improvement as a means of resilience. Completing education and obtaining jobs for survival, going overseas for higher studies, and a desire to learn more about gender and sexuality were among the examples of the research participants’ approaches to self-improvement. Continuity in self-improvement was shown to be a skill for coping shared among the participants as it may keep them engaged in productive pursuits, helping them achieve life goals while aiding them in forgetting and overcoming ridicule and trauma related to gender identity and CSA.

Individuals with adverse childhood experiences require assistance to recover, yet some individuals find meaning in their tragedy that enables them to cope. Despite the well-documented negative effects of childhood experiences, some individuals remain resilient [51]. Interpreting experiences, gaining insight from situations, and attempting to draw strength from tragedy are other approaches that can be taken to become resilient. This was clear from how the research participants managed to tie the idea of turning tragedy into strength into their stories. Different strengths, particularly those connected to self-regulation, meaning-making, and the interpersonal context, contribute to well-being and surviving adversity [51]. The individuals in this study demonstrated these types of strengths while achieving resilience. A series of sexual abuses and many adversities, such as poverty, marginalization, and humiliation due to sexuality, had left deep scars on their mental health. Despite that, they gathered the courage to heal and motivated themselves to work hard, which is an example of self-regulation [51]. Meaning-making measures how people find fulfilment by connecting to something bigger than themselves [51]. The research participants’ attempt to help others and find happiness in seeing happiness in others is an example of meaning-making for well-being. Not only that, but also, when individuals recognize their own experiences and seek ways to support others facing similar hardships—both logistically and mentally—it demonstrates their positive interpersonal skills. The research participants, drawing from their own experiences of abuse, proposed educating children about CSA, exemplifying these skills. This indicates their path to recovery and shows how vulnerability and resilience are tied together.

Practicing extracurricular activities such as reading, artistic activities, etc. can boost resilience. As a preventive factor, a favorable link exists between the number of hours devoted to organised artistic activities and a reduction in the symptoms of depression [52].

Apart from these, family support was also identified as vital, offering a foundation for healing and empowerment [53]. It is possible that many will not receive support from their families when circumstances are challenging, such as being bullied or experiencing CSA. For survivors to cope, it is necessary to consider the importance of family support, parent-child relationships, and parental awareness of childcare. Active coping also requires mental support, counselling, and support from peers and the community. Social networks indeed can promote the establishment of a psychosocial support system in which individuals may express and share their feelings and affection, thereby enhancing their mental and physical health [52]. In this study, all participants notably recognized the importance of family support to cope with adversities, whether it is related to CSA or gender identity. A supportive family can contribute to mental health recovery from bullying, protect children from adversity, and be a solid support for CSA survivor children. A loving family not only helps to heal psychological hardships but may also offer individuals the strength to think creatively, focus on their aspirations, and step outside societal norms.

These research findings regarding resilience can be utilized in future research or interventions to assist gender non-conforming CSA survivors with coping with adversity and their physical, psychological, and emotional well-being. Future interventions should prioritize enhancing mental health support services tailored to the unique needs of transgender CSA survivors.

### 4.3. Advocacy and Social Change for Health Promotion

The participants in this study actively engaged in volunteer work and advocacy, which not only provided them with a sense of purpose but also contributed to their personal healing and social integration [52].

Volunteerism, described by assistance, cooperation, and sympathy, distinguishes pro-social behavior where there is a correlation between volunteer activities and life satisfaction, with volunteers conducting social assistance [54], which is expected to bring resilience. Volunteering was identified as a significant contributor to well-being, offering participants opportunities for socialization and support. According to the research participants, volunteerism and a progressive mindset towards humanity lead to mental peace. They agreed to the fact that their volunteer activities were places where they found social acceptance, promoting their well-being. Perhaps, they viewed volunteer activities as an opportunity for socialization for themselves, who have been marginalized and socially isolated due to their gender identity. Research suggests that volunteering enhances self-efficacy and provides marginalized individuals with a platform for societal engagement and self-expression [54].

The participants’ narratives related to advocacy were aligned with their well-being. This is perhaps due to external social consequences, such as social acceptance that occurs from conformity with the prevalent social norms [54]. Supporting people affected by humiliation can also be considered a crucial coping mechanism, particularly among those (e.g., research participants) who have been victimized in the same way in the past. Though there is a correlation between having appreciation and the value placed on helping others [55], it is a matter of self-satisfaction and mental calmness, according to the research participants. This illustrated how promoting the development of marginalized individuals benefits their own well-being and how advocacy and self-sufficiency are interconnected. The research participants took life’s challenges seriously to attain their goals and advocate for marginal communities. Advocacy efforts indeed serve as powerful tools for fostering resilience and improving social inclusion, highlighting the need for greater community engagement to support marginalized populations [56]. As such, fostering advocacy initiatives and community-based support programs can play a pivotal role in promoting health equity for transgender individuals.

## 5. Conclusions

By examining the life story narratives of four transgender women in Bangladesh, this study sheds light on the complex realities faced by this marginalized group. The findings illustrate the profound impact of societal and cultural barriers on their experiences of CSA and highlight the resilience they demonstrate through coping mechanisms such as self-improvement and advocacy.

Addressing the challenges faced by transgender CSA survivors requires targeted interventions that promote social inclusion, gender-sensitive education, and access to healthcare services. Strengthening support systems at the family and community levels can help facilitate disclosure, reduce stigma, and enhance resilience. Future research should focus on developing culturally appropriate interventions that empower transgender individuals and address the systemic barriers that perpetuate their vulnerabilities.

## Data Availability

The datasets presented in this article are not readily available due to privacy and ethical restrictions, as the interview data contained information about the sensitive life events of the research participants.

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
