# Peer review of "Resilience amid Adversity: A Qualitative Narrative Study of Childhood Sexual Abuse Among Bangladeshi Transgender Individuals"

_ijerph, 2025, doi:10.3390/ijerph22040615_

Round 1
Reviewer 1 Report
Comments and Suggestions for Authors
I appreciated the efforts of the authors to present the stories of these women in a respectful manner to help highlight their life experiences and the impacts of those experiences on their overall health and well-being. At the same time, I wanted to see some more reference to health, as some of the statements provided seemed sweeping and general. It would have been more useful to share some information on exactly how their health was impacted.
I would like to see definitions of resilience included in the literature review, as that seemed like a major gap. Since the authors rely so much on resilience as a base for their work, it needs more attention.
Additionally, I think much of what was discussed in terms of findings/results could also be explained by post-traumatic growth, which was not addressed. In particular, post-traumatic growth can explain a lot of the coping mechanisms described in these stories. Not addressing it at all seems like a major gap.
I was also concerned that the methods section did not contain positionality statements about the authors. I think for this type of sensitive research, especially, authors need to be clear about who they are and how they approached the topic.
Other things missing from the methods that I think need to be addressed include a brief statement, probably following the discussion of adhering to WHO requirements, about receiving IRB approval. Additionally, I wanted more information about how the narratives were condensed and how the authors made decisions about what to include or exclude. Then, when I got the narratives themselves, I wanted to know if those were the summaries referred to earlier in the paper, since that was not clear. I also wanted to know what language the interviews were conducted in as well as who then did the translation and if the translations were verified by anyone.
The reference to family supports was confusing to me, as most of the narratives highlighted a lack of family supports. The authors may consider clarifying some of these statements to support their work.
Below are some other notes I made as I read of errors I found that I think need to be corrected:
1) p. 2, Line 88, seems to be missing a word, as I do not understand what is being stated.
2) p. 3, lines 102-103, sentence construction is not congruent
3) p. 4, line 152, missing letter in the word "was" (ws).
4) p. 5, line 233, should read "ages" instead of "eages"
5) p. 5, line 236, error in spacing
6) p. 6, need a space between the final narrative and the next paragraph
7) p. 7, line 289 should read "was" not "as"
In conclusion, I think this is important work, but addressing some of the concerns above would improve the paper overall.
Author Response
Responses to Reviewer
We thank Reviewer 1 for their thoughtful and constructive feedback, which has helped us to strengthen our manuscript. Below, we outline how each comment has been addressed in the revised version.
- Respectful representation of participants and clearer health references
Reviewer comment: "I appreciated the efforts... but some statements seemed sweeping and general. It would have been more useful to share some information on exactly how their health was impacted."
Response: Thank you. We revised the narratives and the discussion section to include more explicit references to the health impacts of CSA, including mentions of physical injuries, trauma, PTSD-like symptoms, and barriers to accessing healthcare. We aimed for clarity and precision and avoided generalisations when discussing health outcomes. See edits on page 10 lines [19-21].
- Definition and literature on resilience
Reviewer comment: "I would like to see definitions of resilience included in the literature review..."
Response: We agree and have added a clear definition of resilience in the introduction [Page 3 lines 23-25], supported by relevant literature. We also expanded the discussion of resilience in both the results and discussion sections [refer to page 5 lines 22-24, page 11 line 39. Page 12 line 1, 6-40, page 13 line 1-11].
- Integration of post-traumatic growth (PTG)
Reviewer comment: "Much of what was discussed... could also be explained by post-traumatic growth..."
Response: We appreciated this suggestion and have now incorporated a discussion of PTG in both the results and discussion sections. We highlight how participants’ coping strategies and advocacy efforts align with the PTG framework and have added relevant citations. Page 8, lines 25-32, page 9 lines 3-4
- Author positionality
Reviewer comment: "The methods section did not contain positionality statements about the authors."
Response: A brief positionality statement has been added to the methods section, clarifying the authors’ backgrounds in global health, gender-based violence, medical anthropology, and trauma-informed research. Page 37-38, page 4 line 1-3
- IRB and ethical approval
Reviewer comment: "...a brief statement... about receiving IRB approval..."
Response: We have now included a statement on ethical approval and registration with the Norwegian Centre for Research Data and the Ethics Review Committee at the University of Rajshahi. This information is provided in both the methods and Institutional Review Board Statement sections. Page 5 lines 17-20
- Clarification of narrative summaries, translation, and language
Reviewer comment: "I wanted more information about how the narratives were condensed... what language... who did the translation..."
Response: The methods section has been expanded to describe the process of condensing the narratives, how selections were made to align with study aims, and how summaries were validated by participants. (Page 4, lines 31-33, page 4 line 36 to page 5 line 12). We also clarified that interviews were conducted in Bangla, translated by the first author, and reviewed by participants for accuracy. Page 4 lines 23-30
- Clarifying references to family support
Reviewer comment: "The reference to family supports was confusing..."
Response: We revised the text to clarify that although most participants lacked family support, they still emphasised its importance for healing and resilience. This distinction is now more clearly articulated in the discussion section. (Page 12 line 39 to page 13 line 11).
- Minor edits and typographical errors
Reviewer comment: Listed several line-specific typographical errors.
Response: Thank you for identifying these. All noted typographical and grammatical errors have been corrected in the revised manuscript.
We appreciate your insights and feedback that has helped enhance the clarity, depth, and rigour of our manuscript. We hope the revised version now meets your expectations.
Sincerely,
Jennifer J. Infanti and Sanzida Yeasmin
Reviewer 2 Report
Comments and Suggestions for Authors
The manuscript presents an engaging and well-structured study that offers valuable insights into CSA. The introduction effectively contextualizes the research within the existing literature, and the study’s objectives are clearly defined. The writing is clear and well structured.
However, there are some areas that require further improvement to enhance the rigor and clarity of the manuscript. Specifically, the results section primarily consists of narratives without presenting any analytical framework. For the results, consider highlighting key patterns, themes, or categories derived from the data to provide a clearer understanding of how conclusions were drawn. Also, providing illustrative excerpts from participants (if applicable) alongside analytical interpretations would enhance clarity. To strengthen the contribution of this research, a more detailed methodological description of the data analysis process is necessary. Additionally, the discussion should better integrate the findings with previous scientific literature. Rather than reiterating what has already been stated, the discussion should focus on how the findings contribute to, challenge, or extend prior research.
Author Response
Responses to Reviewer 2
We sincerely thank Reviewer 2 for their positive and thoughtful feedback. We appreciate your recognition of the clarity, structure, and contribution of our study. Below, we address the key points raised.
- Presentation of results through thematic categories and excerpts
Reviewer comment: "The results section primarily consists of narratives without presenting any analytical framework... consider highlighting key patterns, themes, or categories... and providing illustrative excerpts."
Response: We appreciate this suggestion and recognise the value of thematic frameworks and illustrative excerpts in qualitative research. However, we intentionally chose a narrative life story approach to preserve the coherence and richness of each participant’s lived experience. This decision aligns with our methodological approach and with the sensitive nature of the topic.
Rather than fragmenting the stories into coded categories and thematic sub-sections, we sought to centre participants’ voices through holistic narrative summaries that reflect their unique trajectories, in line with narrative research principles. Analytical interpretation of shared patterns across narratives is provided in the discussion section, where we draw out key themes related to coping, resilience, and advocacy, supported by literature.
- Further detail on data analysis
Reviewer comment: "A more detailed methodological description of the data analysis process is necessary."
Response: Thank you for this suggestion. We expanded the Materials and Methods section to provide a more detailed account of our thematic narrative analysis process. This includes inductive coding, theme generation, supervision by the co-author, transcript rechecking, and participant validation of narrative summaries. These revisions aim to clarify how analytical conclusions were developed (Page 4, lines 31-33, page 4 line 36 to page 5 line 12).
- Integration with prior research in the discussion
Reviewer comment: "The discussion should better integrate the findings with previous scientific literature..."
Response: We agree and have revised the discussion section to more fully contextualise our findings within existing literature. We now engage more directly with research on CSA, gender identity, disclosure, resilience, and PTG, highlighting both similarities and new contributions from our study. (Page 9 to page 13)
Once again, we thank you for your valuable insights. Your comments have helped us to refine and strengthen the manuscript. We hope the revised version now meets your expectations while maintaining coherence with our chosen methodology.
Sincerely,
Jennifer J. Infanti and Sanzida Yeasmin
Reviewer 3 Report
Comments and Suggestions for Authors
This is what seems to be a very good first take on the material collected by the authors. The material is interesting and comes from the Global South to highlight a set of challenges experienced by trans* individuals based on their adverse childhood experiences. Potentially, it can turn into a very valuable contribution to the field. HOwever, there are some areas that need inmprovement:
1) background - the authors shall provide a more detailed and thorough background on Bangladesh and the status of trans rights there: analysis of legislation (on sexual abuse, trans rights, health issues etc.), de facto situation (NGO reports, research analytics); current debates and conclusions as to why and how adverse childhood experiences in Bangladesh might contribute to better understanding of trans* rights, GBV etc.;
2) I would like to see a more detailed analysis of the ground theory (if any) used or other theoretical and methodological frameworks. So far the description is very basic and does not really provide any understanding or academic premises for analysis;
3) I do not see why full-texts of interviews are included (see above - the justification needs to be provided in the methods section) - the interviews shall be in the appendix and the Results section focussed on findings;
4) the Discussion section is very schematic and does not really provide a deep discussion of the findings. Moreover, any of the cited items can be easily found in any other locality or country. The main question is what is specific about this group of individuals? What are their particular adverse experiences connected to their culture, locality, socio-economic statues etc. How do these experience read against other findings and research? What do they add to our knowledge of adverse experiences of trans* community?
Author Response
We would like to thank Reviewer 3 for their thoughtful and constructive feedback. We greatly appreciate your recognition of the study’s potential contribution and have carefully considered each of your comments. Below, we explain how we addressed the concerns raised.
- Strengthen background context on Bangladesh and trans rights
We have expanded the introduction to include a more comprehensive overview of the sociocultural, legal, and policy context in Bangladesh in relation to transgender rights and childhood sexual abuse (CSA). This includes references to national legislation (e.g., Nari o Shishu Nirjaton Daman Ain 2020, the Domestic Violence Act 2010, and Section 377 of the Penal Code), the absence of anti-discrimination protections for sexual and gender minorities, and the role of societal stigma. We also draw on recent studies and NGO reports to describe the de facto challenges faced by transgender individuals in accessing health and legal services. (Page 2 lines 8-15; page 2 lines 17-26; Page 2 lines 34-41; page 3 lines 1-7) - Clarify theoretical or methodological frameworks
We have elaborated on the theoretical framing of resilience and post-traumatic growth (PTG) as central concepts informing our interpretation of participants’ experiences and strategies. These frameworks now structure the discussion and are integrated into our analysis of coping and advocacy. In addition, we have strengthened the explanation of our narrative methodology and thematic analysis in the Methods section, citing [Braun & Clarke] and other relevant sources. This provides greater clarity about the inductive coding process and how themes were developed. (Page 4, lines 31-33, page 4 line 36 to page 5 line 12). - Explain rationale for presenting life stories in full
Thank you for raising this point. While we understand that qualitative manuscripts often present findings through excerpts, our study employs a narrative approach that centres the holistic life experiences of participants. The summaries presented in the Results section reflect this method and were co-constructed with participants, who reviewed them for accuracy and authenticity. This approach was chosen to preserve narrative integrity and minimise fragmentation. We have clarified our rationale in the Methods section. - Expand and deepen the discussion section
We took this comment seriously and have significantly revised the discussion. The revised discussion is now structured around three overarching themes: (1) gender identity and societal norms; (2) trauma and resilience; and (3) advocacy and social change. We connect our findings to relevant literature from both Bangladesh and international contexts, and reflect on how participants’ experiences offer unique insights into CSA, gender identity, and structural marginalisation in a low-income, conservative setting. While the discussion is now longer, we believe it provides the depth and cultural specificity the reviewer called for.
We hope these revisions address your concerns and enhance the scholarly contribution of the manuscript. Your feedback has been instrumental in improving the depth and clarity of our work.
Sincerely,
Sanzida Yeasmin and Jennifer J. Infanti
Round 2
Reviewer 2 Report
Comments and Suggestions for Authors
After a thorough review of the revised manuscript, I am pleased to confirm that the previously identified areas for improvement have been successfully addressed. The authors have demonstrated a clear commitment to refining their work by incorporating the suggested modifications, strengthening the coherence of their arguments, enhancing methodological rigor, and clarifying key points where necessary. These revisions have significantly improved the manuscript’s overall quality.
Reviewer 3 Report
Comments and Suggestions for Authors
This is much better now!